# Determining the Benzo[a]pyrene Degradation, Tolerance, and Adsorption Mechanisms of Kefir-Derived Bacterium *Bacillus mojavensis* TC-5

**DOI:** 10.3390/foods14152727

**Published:** 2025-08-04

**Authors:** Zhixian Duo, Haohao Li, Zeyu Wang, Zhiwei Zhang, Zhuonan Yang, Aofei Jin, Minwei Zhang, Rui Zhang, Yanan Qin

**Affiliations:** 1College of Smart Agriculture (Research Institute), Xinjiang University, Urumqi 830017, China; 18109998589@163.com (Z.D.); 17393892415@163.com (H.L.); 18205340055@163.com (Z.W.); zhangmw@xju.edu.cn (M.Z.); 2Xinjiang Key Laboratory of Biological Resources and Genetic Engineering, Xinjiang University, Urumqi 830046, China; 3Institute of Materia Medica, Xinjiang University, Urumqi 830046, China; zhangzzw@xju.edu.cn; 4Xinjiang Key Laboratory of Special Species Conservation and Regulatory Biology, College of Life Sciences, Xinjiang Normal University, Urumqi 830054, China; 13720112270@163.com (Z.Y.); 15238780833@163.com (A.J.)

**Keywords:** *Bacillus mojavensis*, kefir, benzo[a]pyrene, degradation pathway

## Abstract

Microbial detoxification, as an environmentally friendly strategy, has been widely applied for benzo[a]pyrene (BaP) degradation. Within this approach, food-derived microbial strains offer unique advantages in safety, specificity, and sustainability for detoxifying food-borne BaP. In this study, we aimed to explore the potential of such strains in BaP degradation. *Bacillus mojavensis* TC-5, a strain that degrades BaP, was isolated from kefir grains. Surprisingly, 12 genes encoding dehydrogenases, synthases, and oxygenases, including *betB*, *fabHB*, *qdoI*, *cdoA*, and *bioI*, which are related to BaP degradation, were up-regulated by 2.01-fold to 4.52-fold in TC-5. Two potential degradation pathways were deduced. In pathway I, dioxygenase, betaine aldehyde dehydrogenase, and beta-ketoacyl-ACP synthase III FabHB act sequentially on BaP to form 4H-pyran-4-one,2,3-dihydro-3,5-dihydroxy-6-methyl via the phthalic acid pathway. In the presence of the cytochrome P450 enzyme, BaP progressively mediates ring cleavage via the anthracene pathway, eventually forming 3-methyl-5-propylnonane in pathway II. Notably, TC-5 achieved an impressive BaP removal efficiency of up to 63.94%, with a degradation efficiency of 32.89%. These results suggest that TC-5 has significant potential for application in addressing food-borne BaP contamination. Moreover, our findings expand the application possibilities of Xinjiang fermented milk products and add to the available green strategies for BaP degradation in food systems.

## 1. Introduction

Benzo[a]pyrene (BaP) acts as a common polycyclic aromatic hydrocarbon (PAH). Inappropriate processing such as high-temperature heating allows BaP to accumulate in edible oil, meat products, dairy products, and beverages [1,2,3]; thus, diet is the main source of BaP for individuals not directly exposed to BaP. Excessive BaP in food is potentially carcinogenic to humans [4,5]. The recalcitrant nature of BaP makes the exploration of efficient and sustainable removal methods a critical challenge in food safety.

A range of techniques is available for the removal of BaP, encompassing chemical oxidation, electrokinetic remediation, and microbial degradation [6]. Chemical oxidation and electrodynamic remediation methods, which involve the addition of high-concentration oxidants or chemicals, have several drawbacks. They can disrupt the diversity of microorganisms, lead to secondary pollution, and entail high operational costs. In contrast, microbial degradation strategies have become increasingly prominent. These strategies work by enzymatically converting pollutants into non-toxic metabolites. As a result, they offer inherent safety and are cost-effective [6]. Bacterial strains originating from coking wastewater, petroleum-contaminated soils, etc., have been conducted to the level of metabolites and related functional genes and enzymes. Structural information on bacterial action on BaP degradation products was first obtained in 1975, when a mutant strain of *Beijerinckia* was able to oxidize BaP to cis-9,10-dihydrodiol benzo[a]pyrene [7]. Bacterial degradation of BaP mainly occurs through enzymatic degradation, secreting degradative enzymes with degradation functions, such as oxidoreductases, hydrolases, and synthetases [8,9,10,11]. Dioxygenases and monooxygenases encoded by the genes *catE*, *Rhd1*, *C23O*, and *CYP102*(*HN14*) can first introduce two or one oxygen atoms into the aromatic ring of BaP to form cis-dihydrodiols and trans-dihydrodiols [11,12]. Subsequently, these products undergo further transformation mediated by intermediates like phenanthrene, pyrene, and naphthalene [9,10,11]. Finally, intermediate metabolites such as low molecular weight catechols, phthalates, salicylic acid, and oxalic acid are then produced via, for example, the salicylic acid pathway of interstitial cleavage of catechols and the phthalic acid pathway [11,12,13]. Moreover, transporters encoded by genes such as *tbdt-11* and *tbdt-23* are involved in the process of BaP transport [14,15,16].

However, contaminating strains are not typically employed for BaP contamination mitigation in food matrices. Characterizing the degradation potential of food-derived microbial strains thus assumes critical importance. Excellent BaP removal was demonstrated by *Lactobacillus plantarum* CICC 22135 and *Lactobacillus pentosus* CICC 23163, derived from fermented foods, and both of them scavenged more than 65% of BaP [17]. *Bacillus velezensis* PMC10 isolated from fermented foods degraded 10 mg/L BaP concentration [18]. These findings on the degradation capabilities of food-derived microbial strains highlight their potential as natural agents for BaP degradation; yet, there is a lack of understanding regarding their underlying degradation mechanisms. Therefore, the intrinsic mechanisms of BaP degradation by food-derived strains could be explored using multi-omics techniques [10,13,19,20,21].

Consequently, the present study aimed to isolate and identify a BaP-degrading bacterial strain from Xinjiang kefir grains, optimize its degradation performance, and explore the underlying degradation mechanisms. Firstly, strain culture conditions were optimized to enhance degradation. Secondly, a series of BaP degradation-related genes were identified through whole-genome sequencing and transcriptomics. Finally, the BaP degradation products were authenticated by GC-MS and analyzed in conjunction with multi-omics techniques to predict unique degradation pathways in food-derived bacteria. This investigation offers a basis for the development and utilization of microbial resources in kefir and creates a scientific argument for the application of food-derived strains for the degradation of food-borne BaP.

## 2. Materials and Methods

### 2.1. Chemical

BaP (purity ≥ 99.7%, HPCL) was obtained from Shanghai McLean Biochemical Technology Co., Ltd. (Shanghai, China). Chromatography-grade methanol, chromatography-grade dichloromethane, KH_2_PO_4_, CH_3_COONa, glucose, MgSO_4_·7H_2_O, MnSO_4_·4H_2_O, Tween 80 were obtained from Xinbo Te Chemical Co., Ltd. (Tianjing, China). Peptone was bought from Aoboxing Bio-Tech Co., Ltd. (Beijing, China). Beef extract powder was purchased from Best Biotech Co., Ltd. (Hangzhou, China). Yeast extract fermentation was bought from Solarbio Technology Co., Ltd. (Beijing, China). Ammonium citrate dibasic (C_6_H_14_N_2_O_7_) was procured from Fuchen Chemical Reagent Co., Ltd. (Tianjing, China). TIANamp Bacteria DNA Kit and 2 × TaqPCRMix were bought from TIANGEN BIOTECH CO., Ltd. (Beijing, China).

### 2.2. BaP-Degrading Bacteria Screen and Identification

All strains were obtained from kefir grains in Tajik Autonomous County of Taxkorgan, Kashgar Prefecture, Xinjiang. The MRS media used for BaP degradation are provided in the Appendix A. BaP was dissolved with acetone.

In previous experiments, tolerance tests to BaP were performed on bacteria screened from kefir grains. The culture medium was 40 mg/L BaP-MRS, and the bacteria were cultured in an oscillatory incubator (HZP-25, Yuejin, Shanghai, China) at 37 °C with shaking (140 rpm) in the dark. The OD_600_ values of culture were recorded by an enzyme-labeled instrument (Multiskan Sky with Touch Screen, Thermo, Waltham, MA, USA) every 4 h for 36 h to draw growth curves of BaP-tolerant strains. The strain with the highest degradation potential was chosen for subsequent analysis based on the OD_600_ value. Follow up strain was identified morphologically and biochemically by the 16S rRNA sequencing technique, gram stain, and SEM (HITACHI-SU8600, Shanghai, China).

### 2.3. BaP Removal Analysis

#### 2.3.1. HPLC Analysis of BaP Biodegradation

After incubation, the sample to be tested was removed and the same volume of chromatography-grade dichloromethane was added. The mixture was vortexed for 30 s, thoroughly mixed, extracted by ultrasonication at 40 °C for 10 min, and centrifuged at 12,000 r/min for 8 min. Then, the lower organic phase was collected. The samples were filtered through an organic phase filter membrane with a pore size of 0.22 μm and stored at 4 °C to be measured. The content of BaP in the solution to be tested was determined by an external standard method using high performance liquid chromatography (HPLC) (Agilent 1260, Santa Clara, CA, USA). The reagent blank group and sample blank control group were installed in this test, respectively. Details of the determination method are given in the Appendix A. The linear equation of the standard curve was obtained as y = 80.5036X + 25.4214 with a correlation coefficient R^2^ = 0.9996 (Appendix A). The *degradation efficiency* was determined according to the formula below [22]:(1)$$Degradation\;efficiency\;\left(\%\right)=\left(C_{0}-C_{t}\right)/C_{0}\times 100\%,$$
where *C_0_* and *C_t_* are the control and treatment groups BaP concentrations, respectively.

#### 2.3.2. Optimization of the Degradation Performance of BaP-Degrading Bacteria

Using the degradation efficiency as a key indicator, single-factor experiments were conducted for three factors: BaP concentration, pH value, and rotational speed. The BaP concentration was 10, 15, 20, 25, and 30 mg/L; the pH was 4, 5, 6, 7, and 8; and the rotational speed was 120, 140, 160, 180, and 200 rpm (Appendix A).

Based on single-factor experiments, three levels for each of the three factors were selected to design an orthogonal experiment. Experiments were conducted using the L_9_(3^3^) orthogonal table to optimize the degradation efficiency (Appendix A). Each treatment was triple-parallel.

#### 2.3.3. HPLC Analysis for BaP Removal

The optimized samples were centrifuged at 12,000 r/min for 8 min to remove bacterial cells. An equal volume of chromatographic-grade dichloromethane was added to the supernatant to extract BaP. The mixture was vortexed for 30 s to ensure thorough mixing, followed by ultrasonic extraction at 40 °C for 10 min. It was then centrifuged again at 12,000 r/min for 8 min, after which the lower organic phase was collected. The organic phase was filtered through a 0.22 μm pore-size organic phase filter membrane. Details of the HPLC conditions are provided in the Appendix A. The *removal efficiency* was calculated according to the following formula [23]:(2)$$Removal\;efficiency\;\left(\%\right)=\left(C_{1}-C_{2}\right)/C_{1},$$
where *C_1_* and *C_2_* are the BaP concentrations of the control and treated groups of samples in the optimal *degradation efficiency* group, respectively.

Finally, the *removal efficiency* is the sum of the degradation and adsorption efficiency of BaP by the strain, and the *adsorption efficiency* can be obtained according to the following formula [23]:(3)$$Adsorption\;efficiency\;\left(\%\right)=Removal\;efficiency\;\left(\%\right)-Degradation\;rate\left(\%\right).$$

### 2.4. Whole Genome Sequencing of Bacillus mojavensis TC-5

The bacterial suspension was centrifuged, and the precipitate was collected (14,000× *g*, 5 min). The bacterial precipitate samples were subsequently sent to the sequencing company (Shaanxi IrunBio Co., Ltd., Xi’an, China) using dry ice after being placed in a −80 °C refrigerator. High-quality DNA was extracted with the Qiagen kit (Shanghai, China) to construct a 1D library, and single-molecule sequencing of DNA was performed using sequencer PromethION (Oxford Nanopore Technologies, Oxford, UK) to obtain raw sequencing data. Low-quality and short-length reads were first filtered out, and the filtered reads were subsequently assembled using Unicycler v0.4.8 software (Ryan Wick, Brisbane, QLD, Australia). For the prediction of tRNAs in the genome, tRNAscan-SE 2.0 software was employed (http://lowelab.ucsc.edu/tRNAscan-SE/ (accessed on 26 March 2024), while the prediction of rRNAs was accomplished with the aid of Barrnap 0.7 software. The presence of genomic islands in the genome was predicted via the IslandViewer 4 website. Meanwhile, the sequences of each gene were annotated using BLAST database (https://blast.ncbi.nlm.nih.gov/Blast.cgi) (accessed on 26 March 2024) in combination with the COG database (http://www.ncbi.nlm.nih.gov/COG/) (accessed on 26 March 2024) and KEGG database (http://www.genome.jp/kegg/) (accessed on 26 March 2024) to obtain genomic annotation information. The sequencing data have been submitted to the National Center for Biotechnology Information (NCBI) with the accession number CP151556.

### 2.5. Transcriptomics Analysis

To analyze the variation in transcript levels of bacterial genes under BaP stress, a treatment group (BaP group) with BaP added and a control group (CK group) without BaP were set up. Each treatment was triple-parallel. Bacteria were collected by freezing and centrifugation after incubation under optimal conditions. The precipitate was then rapidly frozen in liquid nitrogen for 5–10 min and transported to the sequencing company (Biomarker Technologies Co., Ltd., Beijing, China) on dry ice. Total RNA was extracted using the Trizol method. Subsequently, the quality-controlled libraries were subjected to PE150 sequencing using the Illumina HiSeq 2500 Platform. Genes were analyzed for differential expression using the difference analysis software edgeR 4.0 (FDR <= 0.05 & |log_2_FC| ≥ 1).

### 2.6. Analysis of Degradation Products Using GC-MS

To assess the metabolites of BaP degradation by bacteria under BaP stress, the same experimental group as the transcription group was set up, but a blank control group with BaP added and no bacterial solution was added. Samples were taken at the end of the incubation. Then, the samples were centrifuged, the supernatant was taken, and an equal volume of chromatographic-grade dichloromethane was added to the BaP extract and its metabolites. The mixture was vortexed and mixed for 30 s. The sample was extracted by ultrasonic extraction at 40 °C for 10 min and centrifuged at 12,000 r/min for 8 min; the lower organic phase was collected and concentrated 10-fold, and then filtered through the organic phase filter membrane with a pore size of 0.22 μm. A full wavelength scan was then performed using GC-MS (Agilent 7890B-5977A, Santa Clara, CA, USA) (see Appendix A).

### 2.7. Data Processing and Statistical

The phylogenetic tree was established from the 16S rRNA gene sequence of strain TC-5 to those of other strains by using the neighbor-joining method of MEGA 7.0 software. Sequence similarity comparison was performed using RAST (https://rast.nmpdr.org/) (accessed on 15 July 2024) and BLAST (https://blast.ncbi.nlm.nih.gov/Blast.cgi) (accessed on 15 July 2024). Analysis of differential metabolites was performed using SPSS 20.0 software (IBM Corporation, Armonk, NY, USA) (*p*-value < 0.05).

## 3. Results and Discussion

### 3.1. Screen and Characterization of BaP-Degrading Bacteria

Strain TC-5, a Gram-positive, non-flagellated bacillus about 26 μm long and 6 μm wide, formed irregularly folded white colonies on MRS plates (Figure 1a–c). The sequence comparison revealed that *Bacillus mojavensis* TC-5 showed the highest similarity of 99.93% with *Bacillus mojavensis* RS-14 (Appendix A). *Bacillus mojavensis* could degrade intermediate metabolites of BaP from exogenous hazardous substances, such as phenanthrene and phthalate esters [24,25]. Scientific reports regarding the isolation of *Bacillus mojavensis* from kefir are relatively scarce. However, as a traditional fermented food rich in complex microbial communities, kefir exhibits diverse and geographically distinct microbial compositions. Existing studies have demonstrated that the genus *Bacillus* constitutes an important component of kefir grains [26,27]. Notably, the isolated TC-5 exhibits high tolerance to BaP and can still grow at a BaP concentration of 40 mg/L (Figure 1d), suggesting that it may be able to utilize BaP as a carbon source. Surprisingly, as calculated by formula (1), the degradation efficiency of TC-5 reached 24.5% after 60 h of exposure to BaP at a concentration of 20 mg/L and a temperature of 37 °C (Figure 1e).

To effectively enhance the ability of strain TC-5 to degrade BaP, the BaP concentration, pH value that affects the growth of strain TC-5 and its ability to bind to BaP, and rotational speed that affects the dissolved oxygen content, were selected as key factors for the single-factor experiments, and the degradation efficiency was used as an indicator [28]. The results indicated that the highest degradation efficiency was achieved at the BaP concentration of 20 mg/L, pH = 6, and rotational speed of 160 rpm (Appendix A). Orthogonal experiments further yielded that the optimal BaP concentration was 25 mg/L, the optimal pH value was 7, and the optimal rotational speed was 180 rpm (Appendix A). Finally, it was verified that the degradation efficiency under this condition was 32.89%, which was an increase of 34.2% over the unoptimized group. In contrast to the food-derived strain *Bacillus velezensis* PMC10, which degraded 10 mg/L of BaP in 20 days, TC-5 has higher concentrations of degraded BaP and shorter degradation times [18].

The ability of strain TC-5 to remove BaP was further analyzed. According to formula (2), the removal efficiency of the optimized strain for BaP was determined to be 63.94%, and based on formula (3), the adsorption efficiency was calculated to be 31.05%. The chromatograms before and after optimization show the results (Appendix A). Notably, compared with the removal efficiency of 66.76% and 64.31% for *Lactobacillus plantarum* CICC 22135 and *Lactobacillus pentosus* CICC 23163 at 10 mg/L, the removal efficiency of strain TC-5 was 63.94%, with a 150% increase in BaP concentration [17].

### 3.2. Genomic Analysis of Bacillus Mojavensis TC-5

To elucidate the underlying molecular mechanisms of BaP degradation by the strain TC-5, whole-genome sequencing and comparative genomic analysis were performed. The genome of TC-5 consists of a circular chromosome without a plasmid, and the size of the genome is 3,940,000 bp, with a GC content of 43.79%, containing 86 tRNAs, 5 rRNAs, and consisting of 3911 coding genes (Appendix A). The genome of TC-5 was annotated with 2400 genes (61.37%) in the KEGG database, which were classified by gene function as six classes. Obviously, the genes involved in metabolism accounted for the highest proportion (Appendix A).

Among above these genes, 82 genes were revealed to be associated with the degradation, adsorption, and tolerance of BaP by TC-5, respectively (Appendix A), and they were distributed according to gene function in the following categories: aromatic compound degradation (ko01220), aminobenzoate degradation (ko00627), fatty acid degradation (ko00071), nitrogen metabolism (ko00910), amino acid biosynthesis (ko01230), carbon metabolism (ko01200), fatty acid metabolism (ko01212), ABC transporters (ko02010), and quorum sensing (ko02024), among others. Interestingly, gene sequence comparison and analysis revealed significant sequence homology between the predicted genes and reported genes functionally characterized in BaP degradation (Appendix A). Among them, the sequences of *fabHB*, *bioI*, *betB*, *dapA*, *gltB*, *cypD,* and *acdA*, encoding synthase, oxygenase, and dehydrogenase, respectively, showed 30.73% to 71.91% sequence similarity to genes known to have BaP degradation functions (Figure 1f and Appendix A). This similarity in gene sequences often implies functional conservation and similarity.

### 3.3. Differentially Expressed Genes of Bacillus mojavensis TC-5 Under BaP Stress

To investigate the gene function of TC-5 under the stress of BaP, the transcriptome of TC-5 was analyzed by RNA sequencing. Biological reproducibility of the generated sequences and expressed genes was assessed by correlation analysis, which showed a good correlation between the samples (correlation coefficient: 0.74–0.99) (Figure 2a). A total of 542 up-regulated and 722 down-regulated genes were identified (Figure 2b). Inter-sample expression analysis yielded 238 genes unique to the BaP group and 3357 genes shared with the CK group (Figure 2c). Further analysis of 82 genes revealed that 29 differentially expressed genes were closely associated with degradation, adsorption, and tolerance of BaP by strain TC-5 (Figure 3).

#### 3.3.1. In-Depth Analysis: Mechanism of BaP Degradation by Strain TC-5

First and foremost, the expression of 12 genes, *fabHB*, *bioI*, *betB*, *qdoI,* and *cdoA*, among others, were found to be up-regulated from 2.01-fold to 4.52-fold (Figure 3 and Table 1). BaP could be ring-opened and oxidized by bacterial oxygenases, cytochrome P450 enzymes, dehydrogenases, etc., and then further degraded to carbon dioxide and water via the TCA cycle [9,29,30]. The monooxygenase and dioxygenase encoded by the genes *qdoI, ssuD*, and *cdoA* were hypothesized to partake in the hydroxylation of BaP. These enzymes could introduce two or one oxygen atoms at different sites in the aromatic ring of BaP to form cis-dihydrodiol or trans-dihydrodiol, respectively [8,30]. And the similarity between the *bioI* gene encoding cytochrome P450 enzyme and the cytochrome P450 enzyme gene of *Bacillus thuringiensis* GIMCC1.817 was found to be 30.73% [28]. It may also be implicated in the hydroxylation process of BaP, with the hydroxylated metabolites subsequently being channeled into the metabolic pathways characteristic of phenanthrene and naphthalene [9,10,28]. Meanwhile, the gene *dapA*, which encodes 4-hydroxy-tetrahydrodipicolinate synthase, displayed a 59.31% sequence similarity to dihydrodipicolinate synthase. This enzyme is primarily responsible for the scission of the C-O bond [31]. Additionally, the xanthine dehydrogenase subunit E, xanthine dehydrogenase subunit D, and betaine aldehyde dehydrogenase encoded by the *pucE*, *pucD*, and *betB* genes in strain TC-5 have sequence similarities of 42.86%, 27.35%, and 31.34% with the dehydrogenase, respectively. Dehydrogenase is involved in the re-aromatization process at the intermediate stage of BaP degradation and could produce phthalates [14,29]. Lastly, the *fabHB* gene encoding the beta-ketoacyl-ACP synthase III FabHB enzyme was annotated in strain TC-5. It is involved in a pathway where the BaP intermediate metabolite, phthalic acid, is converted into protocatechuic acid esters, which are then integrated into the TCA cycle [32]. Based on the above degradation mechanism, the obtained degradation efficiency of 32.89% showed good agreement.

#### 3.3.2. In-Depth Analysis: Mechanisms of Adsorption and Tolerance of BaP by Strain TC-5

In addition to these twelve genes, our study identified four genes (*lytR_1* and *xth*, etc.) that were up-regulated from 2.31 to 2.44-fold (Figure 3). The enzymes encoded by these genes have been implicated in processes such as signal transduction and gene regulation during biofilm generation, which can facilitate BaP adsorption [33,34]. This mechanism is also consistent with the result of a 31.05% adsorption efficiency. There were also nine genes (*appD* and *kdgK*, etc.) down-regulated 0.28-0.41-fold and four genes associated with defense mechanisms (*yadG* and *yxdM*, etc.), up-regulated by a maximum of 58.83-fold in expression (Figure 3). Among them, nine down-regulated genes were associated with the energy transport and pentose phosphorylation pathways [14,15,35]. The above 13 genes together constitute the mechanism of BaP tolerance in the bacterium [36,37], giving strain TC-5 a high tolerance to 40 mg/L BaP. In summary, it was found that the mechanisms of strain TC-5 in response to BaP were degradation, adsorption, and tolerance, respectively (Figure 4).

### 3.4. Intermediate Metabolites and Unique Metabolic Pathways of BaP

To further predict the degradation pathway, intermediate metabolites were determined by GC-MS to investigate the degradation pathway of BaP by strain TC-5. When TC-5 responded to BaP, several types of oxidation products such as alkanes, aromatic organic compounds, esters, etc. were detected, and total ionograms between the different groups are shown in Appendix A. They may be related to BaP degradation, and two major BaP metabolites were detected (3-methyl-5-propylnonane, RT = 12.871 min; 4H-Pyran-4-one,2,3-dihydro-3,5-dihydroxy-6-methyl, RT = 5.71 min) (Appendix A and Table 2). Upon the degradation of BaP, the esterification of phthalic acid substances takes place [38]. The occurrence of esters implies the possible degradation of phthalic acid. However, no phenanthrene, pyrene, and phthalic acid substances were detected during the test, which may be due to the fact that most of the phthalic acid substances have been converted to low molecular weight substances.

Based on the intermediate metabolites, a potential biodegradation pathway for BaP degradation by TC-5 could be hypothesized (Figure 5). Pathway I first involves the initial step of hydroxylation and ring-opening by two oxygenases, quercetin 2,3-dioxygenase (*qdoI*)/cysteine dioxygenase family protein (*cdoA*), whereby the 4,5 and 11,12 carbons of BaP are hydroxylated and ring-opened, which in turn generates 4,5-chrysenedicarboxylate and cis-4-(8-Hydroxypyren-7-yl)-2-oxobut-3enoate. A similar pathway was found in *Microbacterium* sp. M. CSW3 [21]. These compounds then further generate phenanthrene-3,4-diol via the phenanthrene pathway and pyrene pathway, respectively, in a series of ring-opening and transformation processes. Phenanthrene and pyrene have been shown to be intermediate metabolites of BaP [10,11,21,22], and it has been demonstrated that *Bacillus mojavensis* can further degrade phenanthrene [24]. Subsequently, phenanthrene-3,4-diol further opens the ring to form 1-Hydroxy-2-naphthoate. It is then dehydrogenated by betaine aldehyde dehydrogenase (*betB*) to produce phthalate, which enters the phthalic acid pathway. Phthalic acid, as a BaP intermediate [12,22], can be further degraded by *Bacillus mojavensis* [25]. Under the enzymatic action of beta-ketoacyl-ACP synthase III FabHB (*fabHB*), which is capable of degrading phthalic acid [32], and other transformations, the product is 4H-pyran-4-one,2,3-dihydro-3,5-dihydroxy-6-methyl. Phthalic acid can finally be degraded to CO_2_ and H_2_O via the TCA cycle. Pathway II involves the action of cytochrome P450 enzyme (*bioI*) on the 11,12 carbons of BaP, resulting in the generation of benzo[a]pyrene-11,12-epoxide. Through a series of ring-opening and conversion processes, this compound enters the anthracene pathway and produces 1,2-anthracenediol. After further ring-opening, 3-methyl-5-propylnonane was produced. And finally, the product is recycled through the TCA cycle to produce CO_2_ and H_2_O.

The above analyses indicate that there are two pathways for the degradation of BaP by TC-5, namely the phthalic acid pathway and the anthracene pathway. Compared with the reported degrading strains, strain TC-5, derived from edible kefir grains, possesses a unique set of key genes (including *fabHB* and *bioI*) (Appendix A) which are closely associated with its distinct metabolic pathways. Interestingly, 4H-pyran-4-one,2,3-dihydro-3,5-dihydroxy-6-methyl and 3-methyl-5-propylnonane identified in this study have not been found before; thus, we speculated that there are two unique pathways for degradation of BaP by food-derived strains (Appendix A), thereby presenting substantial potential for the degradation of food-borne BaP (Appendix A). Based on the above-defined molecular mechanisms and functional characteristics, this strain has practical application potential in the food industry. Subsequently, this strain is expected to be applied to dairy products as a starter culture, thereby reducing the concentrations of BaP in dairy products and consequently contributing to human health [39].

## 4. Conclusions

In conclusion, *Bacillus mojavensis* TC-5 from kefir grains demonstrates significant potential for the degradation of BaP. The optimized degradation efficiency was 32.89% and the overall removal efficiency was 63.94%. A total of 29 differentially expressed genes that were closely associated with degradation of BaP were identified, and we concluded that the response mechanisms of strain TC-5 to BaP were degradation, adsorption, and tolerance, respectively. Among them, the five degradation genes, *fabHB*, *qdoI*, *cdoA*, *betB*, and *bioI*, are involved in the production of 4H-pyran-4-one, 2,3-dihydro-3,5-dihydroxy-6-methyl, and 3-methyl-5-propylnonane. After that, it was speculated that the degradation pathways of BaP by the food-derived *Bacillus mojavensis* TC-5 were the phthalic acid pathway and the anthracene pathway, respectively. This comprehensive understanding of TC-5’s metabolic BaP capabilities underscores its promising application in the bioremediation of food-borne BaP, enhancing food safety and protecting public health from potential contamination.

## Figures and Tables

**Figure 1 foods-14-02727-f001:**
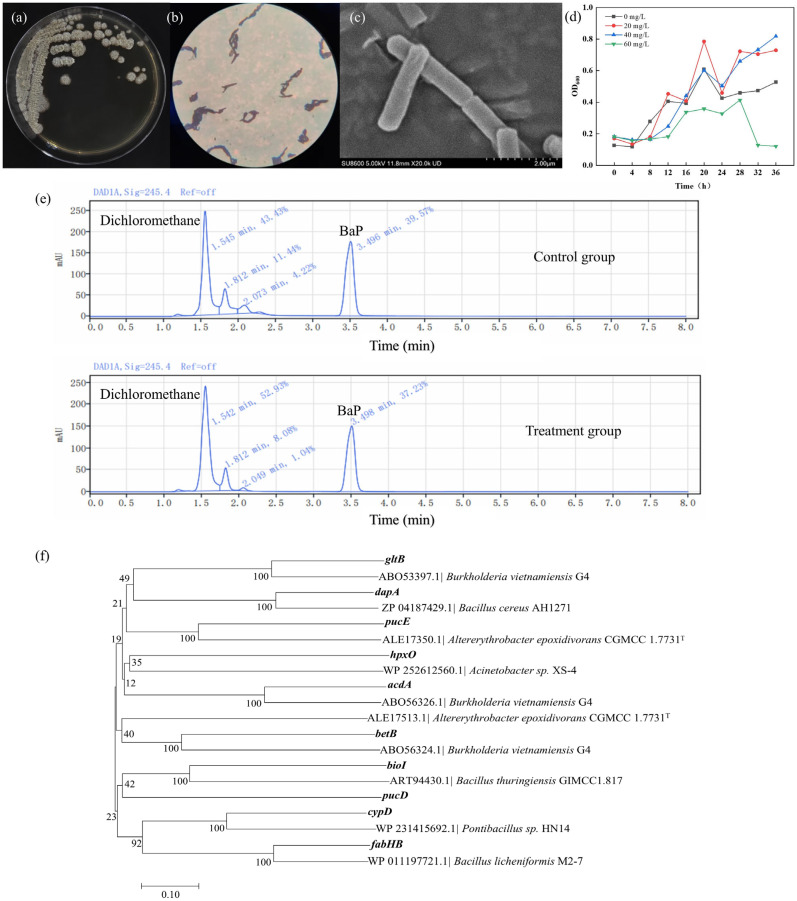
Morphological observations of strain TC-5. (**a**) Colony morphology on plate; (**b**) Gram stain under the microscope; (**c**) external appearance of the bacterium under SEM; (**d**) growth curves of TC-5 at different concentrations of BaP; (**e**) degradation chromatograms of the control and treatment groups; (**f**) phylogenetic tree of genes in TC-5 that may have BaP degradation functions.

**Figure 2 foods-14-02727-f002:**
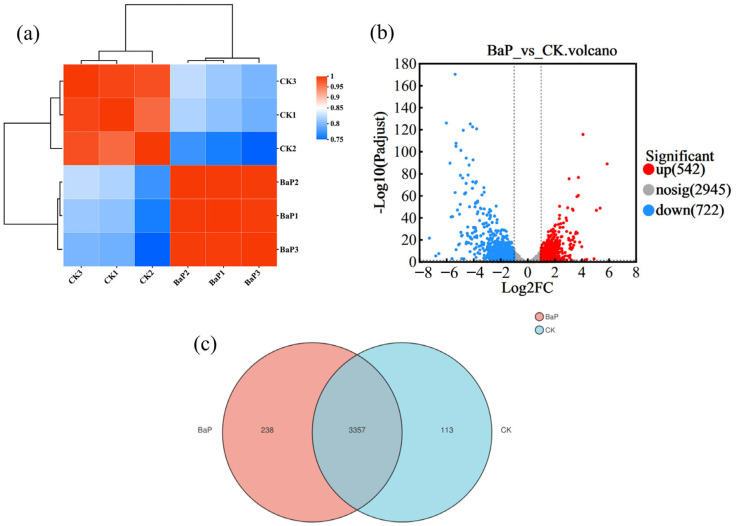
(**a**) Heatmap of correlation analysis between BaP and CK groups; (**b**) volcano map of expressed genes; (**c**) Venn analysis of expressed genes.

**Figure 3 foods-14-02727-f003:**
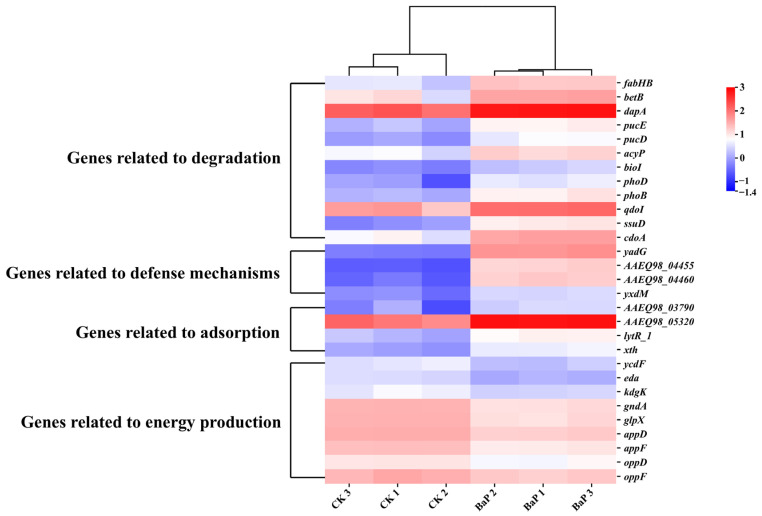
Heatmap of clustering of different functional genes in TC-5.

**Figure 4 foods-14-02727-f004:**
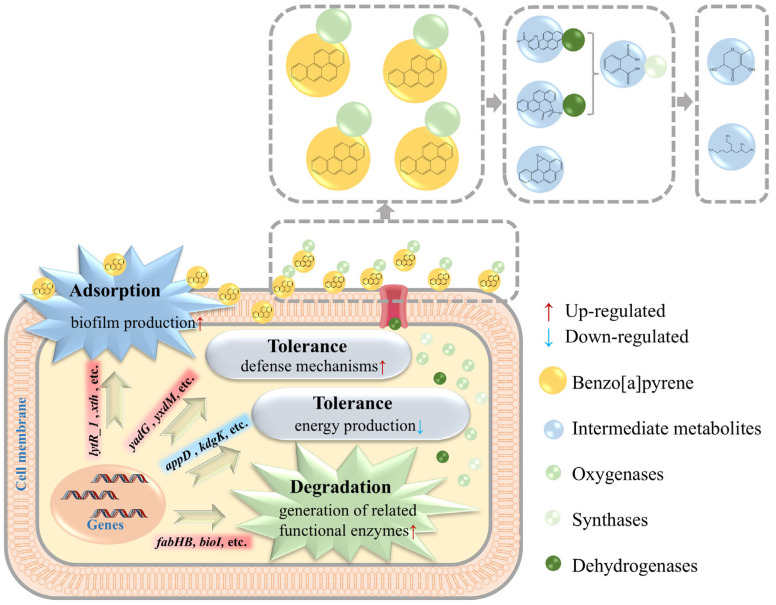
Schematic illustration about mechanism of action of BaP.

**Figure 5 foods-14-02727-f005:**
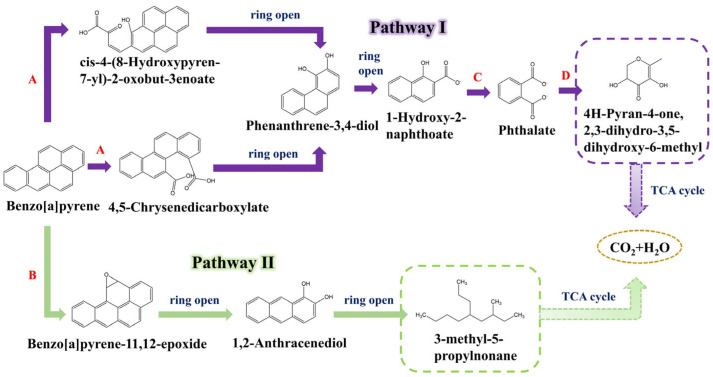
The speculated pathway of BaP degradation by TC-5. A–D represent different enzymes: A: quercetin 2,3-dioxygenase/cysteine dioxygenase family protein; B: cytochrome P450 enzyme; C: betaine aldehyde dehydrogenase; D: beta-ketoacyl-ACP synthase III FabHB.

**Table 1 foods-14-02727-t001:** 12 up-regulated expressed genes involved in BaP degradation.

Gene Name	Accession No.	Predicted Function
*fabHB*	AAEQ98_05520	beta-ketoacyl-ACP synthase III FabHB [EC:2.3.1.180]
*betB*	AAEQ98_15130	betaine aldehyde dehydrogenase [EC:1.2.1.8]
*dapA*	AAEQ98_09030	4-hydroxy-tetrahydrodipicolinate synthase [EC:4.3.3.7]
*pucE*	AAEQ98_15870	xanthine dehydrogenase subunit E [EC:1.17.1.4]
*pucD*	AAEQ98_15875	xanthine dehydrogenase subunit D [EC:1.17.1.4]
*acyP*	AAEQ98_04155	acylp1.3.13.3hosphatase [EC:3.6.1.7]
*bioI*	AAEQ98_14610	cytochrome P450 [EC:1.14.14.46]
*phoD*	AAEQ98_01630	alkaline phosphatase PhoD [EC:3.1.3.1]
*phoB*	AAEQ98_03160	alkaline phosphatase PhoB [EC:3.1.3.1]
*qdoI*	AAEQ98_19870	quercetin 2,3-dioxygenase [EC:1.13.11.24]
*ssuD*	AAEQ98_04825	alkanesulfonate monooxygenase [EC:1.14.14.5 1.14.14.34]
*cdoA*	AAEQ98_15170	cysteine dioxygenase family protein [EC:1.13.11.20]

**Table 2 foods-14-02727-t002:** Degradation product information of BaP.

Name	CAS	RT (min)	Molecular Weight (g/mol)	Molecular Formula	*p*-Value
3-methyl-5-propylnonane	31081-18-2	12.871	184.36	C_13_H_28_	<0.0001
4H-Pyran-4-one,2,3-dihydro-3,5-dihydroxy-6-methyl	28564-83-2	5.71	144.13	C_6_H_8_O_4_	<0.0001

## Data Availability

The original contributions presented in the study are included in the article/Appendix A, further inquiries can be directed to the corresponding author.

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
