# Peer review of "Determining the Benzo[a]pyrene Degradation, Tolerance, and Adsorption Mechanisms of Kefir-Derived Bacterium Bacillus mojavensis TC-5"

_foods, 2025, doi:10.3390/foods14152727_

Round 1

Reviewer 1 Report

Comments and Suggestions for Authors

Manuscript number Foods 3768383

Title: Determining the benzo[a]pyrene degradation, tolerance, 2 and adsorption mechanism of food-derived bacterium Bacillus mojavensis TC-5 from kefir

Here are my suggestions:

Major:

  • It is not clear where the origin of Bacillus mojavensis is. Kefir has several strains of bacteria and yeast. How did the authors select the potential microorganisms that degrade BaP from kefir? How did they perform the screening of the strains?
  • Line 153: Bacteriophage samples were subsequently sent to the sequencing company. What bacteriophages are referring to?
  • I could not find any scientific evidence that correlates Bacillus mojavensis with kefir. Please add information about the isolation and scientific reports from other authors that have isolated this bacterium from kefir.
  • The experimental design is confusing. In general, when the objective is to verify the utilization of a compound, a synthetic medium is formulated, and only this compound is added as a carbon source. In my opinion is not probable that in a medium containing 1% peptone, 1% beef extract, and 2% glucose, the microorganisms utilize the BaP (0,004 %) in any way.
  • It is not valid to compare the BaP tolerance between Bacillus velezensis PMC10 and Bacillus mojavensis, as experiments were performed in different media and conditions.
  • Please identify the peaks of the chromatograms in Figure 1. Where is the BaP? What are the other peaks?
  • How could the application of this strain be used in a real meal? Please speculate about it.

Reviewer 2 Report

Comments and Suggestions for Authors

The article "Determining the benzo[a]pyrene degradation, tolerance, and adsorption mechanism of food-derived bacterium Bacillus mojavensis TC-5 from kefir" presents clear experiments and results. It thoroughly characterizes B. mojavensis TC-5's genetics and its BaP removal mechanisms, using effective graphs and tables to illustrate findings. To enhance clarity, the following recommendations are suggested: 1. It is recommended to number all equations presented for easier reference within the text, allowing readers to locate them easily during discussions or result analyses. 2. Change "degradation rate (%)" to "efficiency of degradation (%)". This ensures that it reflects the percentage of contaminant removed without implying a temporal aspect. Similarly, replace "adsorption rate" with "adsorption efficiency" and "removal rate (%)" with "removal efficiency". 3. Clarify how it was determined that the removal of BaP in Section 2.3.1 occurred predominantly through biodegradation rather than biosorption. If it cannot be definitively shown that the removal is exclusive to biodegradation, state that the observed efficiency reflects an overall removal efficiency for BaP. 4. It is essential to revisit the current definition of BaP removal efficiency via biosorption (currently referred to as "biosorption rate" in the manuscript). To accurately calculate this efficiency, experiments must first be conducted using inactive biomass. This ensures that any observed removal can be attributed solely to the biosorption mechanism, as there will be no metabolic activity. Next, determine biodegradation efficiency by subtracting the amount of BaP removed via biosorption from the total removal observed in active cells under the same conditions. 5. Clearly differentiate between biodegradation and other removal methods described in the manuscript, especially those discussed in Sections 2.3.1 and 2.3.3. This distinction is crucial to avoid confusion regarding experimental conditions and results. If experiments with inactive biomass were conducted and a clear differentiation between degradation and biosorption was achieved, this should be explicitly stated in the corresponding sections of the text.

Reviewer 3 Report

Comments and Suggestions for Authors

The authors studied the degradation of BoP, tolerance, and adsorption by kefir-derived bacterium Bacillus mojavensis TC-5. The study is considered an important research to follow the fate of BoP in an environment dominated by the Bacillus mojavensis TC-5 strain, which can be regarded as a biological method for removing BoP. The study design was performed correctly; sampling and analysis were conducted according to standard methods. The results are described and presented in the manuscript, but the manuscript needs significant revision.

Comments:

  • Title: The title is appropriate, replace “mechanism” with “mechanisms” since three mechanisms were studied. Also, the term “kefir-derived bacterium” can be used in rewriting the title.

  • English Language: The English of the manuscript is very weak and, most of the time, confusing; therefore, it needs significant proofreading.

  • Other comments:
  • Abstract: Line 22- 30: It is not necessary to describe detailed methodology, results and discussions in the Abstract. Just briefly describe the idea behind the study, aims and objectives and the key results obtained, then the details can be described in the main sections of the manuscript.
  • Line 46-47: please rewrite the statement to make it more clear statement.
  • Line 57: “Then continue”, what do you mean by that ? rewrite
  • Lines 74-75: This section needs to be updated to make it easier for the reader to understand the aims and objectives of the study.
  • Line 126: needs a short title, the details can be described in the section, rewrite.
  • Line 130: Do you mean rotational speed or centrifugal speed?
  • Line 141- 152: Can you replace “The organic phase” with “ The supernatant…”
  • Line 152-154: Did the whole bacterial suspension or only the pellets produced from centrifugation of the samples, are sent for sequencing analysis?
  • Line 157-166: very difficult to understand, please rewrite.
  • Line 169: Which Experiment group? Can you use an academic term instead?
  • Line 173-177: only necessary to briefly describe the standard method applied, not the details of the method.
  • Line 178: Can you update the title as “Analysis of degradation products using GC-MS” or “Gas chromatography-mass spectrometryanalysis”
  • Line 367-372: not necessary to repeat the methodology in the Conclusion section; only describe key findings from your study.
  • Which accumulated metabolites from the BoP degradation might cause health risks to humans?
  • Compare /What were the main differences between the BoP degradation by Kefir strains and the other BoP degradation methods described in the literature?
Comments on the Quality of English Language

The English of the manuscript is very weak and, most of the time, confusing; therefore, it needs significant proofreading.

Round 2

Reviewer 3 Report

Comments and Suggestions for Authors

NA